# Floral Plantings in Large-Scale Commercial Agroecosystems Support Both Pollinators and Arthropod Predators

**DOI:** 10.3390/insects12020091

**Published:** 2021-01-21

**Authors:** Eric G. Middleton, Ian V. MacRae, Christopher R. Philips

**Affiliations:** 1Department of Entomology, University of Minnesota, 1980 Folwell Ave, 219 Hodson Hall, Saint Paul, MN 55108, USA; 2Department of Entomology, University of Minnesota Northwest Research and Outreach Center, 2900 University Ave, Crookston, MN 56716, USA; imacrae@umn.edu; 3Department of Entomology, Virginia Tech, 170 Drillfield Drive, 220 Price Hall, Blacksburg, VA 24061, USA; philipsc@iskbc.com

**Keywords:** agroecosystems, pollinators, predators, habitat management, floral plantings, floral resources

## Abstract

**Simple Summary:**

Pollinators and insect predators are in decline, largely due to commercial agricultural land use and practices. Planting a mixture of wildflowers in the unused margins of agricultural fields may help to conserve these insects and the important benefits that they provide (pollination and pest suppression). We compared wildflower plantings around commercial potato fields to unmanaged grass and weed margins to determine whether these plantings supported greater numbers of pollinators and predators. We found that wildflower plantings increased the numbers of both pollinators and predators within field margins. Additionally, margins with more flowers blooming led to more pollinators, although, interestingly, more flowers did not lead to more predators. This suggests that predators may benefit from wildflower plantings without needing the flowers they provide, while pollinators benefit from flowers specifically. When we measured pollinators and predators in the nearby potato crops, we found that wildflower plantings did not lead to greater numbers of pollinators or predators. Our results suggest that wildflower plantings can help conserve pollinators and predators in commercial agricultural areas, but that these beneficial insects do not move into adjacent crops, where they would be most likely to provide pollination or pest suppression services.

**Abstract:**

Beneficial insect populations and the services that they provide are in decline, largely due to agricultural land use and practices. Establishing perennial floral plantings in the unused margins of crop fields can help conserve beneficial pollinators and predators in commercial agroecosystems. We assessed the impacts of floral plantings on both pollinators and arthropod predators when established adjacent to conventionally managed commercial potato fields. Floral plantings significantly increased the abundance of pollinators within floral margins compared with unmanaged margins. Increased floral cover within margins led to significantly greater pollinator abundance as well. The overall abundance of arthropod predators was also significantly increased in floral plantings, although it was unrelated to the amount of floral cover. Within adjacent potato crops, the presence of floral plantings in field margins had no effect on the abundance of pollinators or predators, although higher floral cover in margins did marginally increase in-crop pollinator abundance. Establishing floral plantings of this kind on a large scale in commercial agroecosystems can help conserve both pollinators and predators, but may not increase ecosystem services in nearby crops.

## 1. Introduction

Insect abundance and diversity are decreasing across many taxa worldwide [1,2,3]. While there are many factors affecting this loss, the expansion and intensification of agriculture is a primary driver [4,5]. In particular, commercial, conventionally managed agriculture has been highly disruptive to habitat on which insects rely [6,7,8,9]. 

Beyond being a conservation concern, insects provide important ecosystem services. Within agriculture, biological control and pollination services provided by insect natural enemies and wild bees were estimated to be worth over $7.5 billion in the US [10] and 35% of total crop production worldwide depends on pollinators [11]. As human populations have grown, our reliance on these ecosystem services has also grown [12,13]. Despite this reliance on insect-provided ecosystem services, current agricultural practices lead to a loss of both insect abundance and of ecosystem function [14]. Reductions in insect species richness due to commercial agriculture can severely impact the benefits that humans receive from ecosystems, especially within agricultural settings themselves [15]. Modifying existing agroecosystems to be more sustainable will be beneficial to conservation efforts [16] and may help curb the loss of ecosystem services that benefit agricultural production. 

Establishing plantings of perennial flowers in the unused margins of agricultural fields has been proposed as an effective method to both preserve biodiversity and provide ecosystem services in agricultural settings [17,18]. Floral plantings provide important forage such as nectar and pollen, as well as sources of prey, overwintering habitat, and improved microclimate for various insect taxa [19,20,21,22,23]. 

Floral plantings have been most frequently investigated for their potential to promote pollinators and pollination services. The overall abundance of pollinators and of specific taxa can increase with the presence of wildflower plantings [24,25] with bumblebees in particular responding positively to the addition of floral resources [26,27]. Floral plantings can also promote pollinators within nearby agricultural areas, and have both increased the abundance of wild bees and hoverflies in adjacent blueberry fields [28] and have been linked to greater abundance of wild bees in the surrounding landscape [25]. Pollinators that spill over from floral plantings into fields can provide valuable pollination services to a wide variety of crops [28,29,30]. 

Although much attention has been given to how floral plantings impact pollinators, plantings can also promote insect predators. Floral plantings can attract multiple predatory taxa and increase their abundance compared to control areas [31,32]. These predators can also spill over into adjacent crops and provide biological control services [21,33,34]. In some circumstances, the presence of natural enemies precludes the need for pesticide applications to control crop pests [32]. Since floral plantings in agroecosystems can provide resources for pollinators and predators, several studies have examined if plantings can promote both at once. Wildflower plantings simultaneously increased pollinator and predator abundance [31,35,36], and provided both increased biological control services and pollination services to nearby crops [28,33]. The efficacy of floral plantings can vary across agroecosystems however, largely due to crop type or the surrounding landscape [7], and beneficial taxa can respond differently to plantings [30,37]. Little research has been conducted on the impacts of floral plantings when implemented on farm in commercial, conventionally managed agricultural landscapes, where the losses of insect abundance and diversity are most pronounced. Furthermore, the plantings that have been studied are usually small, often measuring only a few hundred square meters at most. This is particularly important because large floral plantings can have different impacts on the number and type of insects that are attracted compared to smaller plantings [37,38]. To date, few studies have examined the impact of floral plantings on both pollinators and predators in commercial settings, especially for large plantings.

To determine how large, perennial floral plantings impact pollinators and predators in commercial agroecosystems, this study addresses four principal questions:What impact do floral plantings have on pollinators in the margins of commercial agricultural fields?How do floral plantings affect pollinators within agricultural fields?What impact do floral plantings have on predators in the margins of commercial agricultural fields?How do floral plantings affect predators within agricultural fields?

## 2. Materials and Methods 

### 2.1. Experimental Design

Fieldwork was conducted in central Minnesota, where a commercial potato grower established ~2 km^2^ of floral plantings around 42 conventionally managed, center-pivot irrigated potato fields in 2015 and 2016. Floral plantings were created using a commercially available seed mixture marketed by Syngenta and Pheasants Forever [39] consisting of two varieties: a “Honeybee” mixture and a “Monarch” mixture (Appendix A). Floral plantings were seeded in the unmanaged margins of the fields, in sections ranging from 2200 to 20,000 m^2^ depending on the available space. Plantings were established in 1–3 of the four available corners around the potato fields, while the remaining corners were left as unmanaged vegetation (Figure 1).

Fields in the study area were on a 3 year rotation of potatoes, corn, and soybeans or dry beans. To minimize variability due to different crop types, only potato fields were sampled. Potato fields were conventionally managed, with biweekly aerial fungicide applications and an in-furrow neonicotinoid treatment at planting. Potato fields that had at least one corner planted with flowers and at least one corner left unmanaged, and had margins at least 400 m apart were selected for this study. Six fields consisting of seven floral margins and nine control margins in total were sampled in 2017, eight fields with a total of 10 floral margins and eight control margins in 2018, and six fields with six floral margins and six control margins total in 2019. Sampling occurred approximately once a month from late May until late September for a total of five sampling dates per year. In 2018, sampling only took place from May to August due to consistent cold temperatures and rain in September. 

### 2.2. Floral Sampling

In our study system, the grower who established the floral plantings was unsure exactly which fields had been sown with which of the two seed mixtures, or whether the mixtures had been combined. For this reason, we classified all floral plantings as the same for analysis, and measured floral cover to determine how well floral plantings established. 

Floral cover was assessed for both floral and control margins via transects. Starting in the center of each corner margin, and moving towards the crop, a 1 × 1 m square made of PVC tubing was placed at five-meter intervals along a 15 m transect, for a total four sampling sites per margin. All flowers within the square were counted and identified to species. Flowering forbs were identified using the Wildflowers of Minnesota Field Guide [40] and verified in the lab using the website Minnesota Wildflowers [41]. Floral cover was measured by counting the total number of flowers or inflorescences (depending on species) and calculating the average flower or inflorescence area for each species using flower size data available at Minnesota Wildflowers, verified with in-field observation. Area was summed across all species and sampling sites for each field margin, and converted to average percent floral cover per square meter for final analysis. 

### 2.3. Pollinator Sampling

Pollinators were sampled in field margins and inside the potato crops adjacent to each field margin. Sampling occurred at four locations in the crop: the edge (0 m), and 10, 30, and 50 m into the crop. At each location, pollinator sampling was conducted via pollinator surveys, sweep net transects, and incidental catch from pitfall traps placed in pairs. Pollinator surveys consisted of timed walks with an aerial net, capturing all observed bees, hoverflies, and bee flies within 3 m to either side. A timer was started and kept running until a pollinator was captured. At this point, the timer was stopped until the pollinator was transferred to a kill jar, and restarted once the sampler began walking again. Surveys in the margins of fields consisted of 5 min of active sampling time, and surveys at all locations within the potato crop lasted 3 min each. Captured pollinators were frozen for later identification. Sweep-net transects consisted of 50 pendulum sweeps with a heavy canvas sweep net while walking slowly through the vegetation. Pitfall traps were created by burying 180 mL plastic cups (Solo brand, Dart Container Corporation Mason, MI, USA) up to their brim in the ground, and filling them 1/4 full of water with dish soap mixed in. Two pitfall traps were placed ~2 m apart at each sampling location (margin, edge, 10 m, 30 m, 50 m), and were pooled for analysis. Pitfall traps were collected after 18–24 h. All collected insects were frozen for later identification. Syrphidae and Bombyliidae were identified to family, and bees were identified to genus using taxonomic keys on the website Discover Life [42]. Pollinator data from pollinator surveys, sweep nets, and pitfall traps were pooled together, and the total number of pollinators per sampling location per field were summed to estimate overall pollinator abundance both in field margins and within potato crops. The same process was applied to individual taxa.

### 2.4. Predator Sampling

Predators were sampled in field margins and inside the crop at the edge, and 10, 30, and 50 m into the crop. Predator sampling was conducted via sweep net transects and pitfall traps, in the same manner as described for pollinators. Insect predators were identified to the family level, and non-insect arthropod predators identified to the order level, with the exception of the members of the spider families Salticidae and Thomisidae. Predator data were pooled together, and the total number of predators per sampling location per field summed to estimate overall predator abundance. 

Predators were further broken down into categories of “Epigeal” and “Foliar” for analysis. Epigeal predators were those in the insect families Carabidae, Staphylinidae, and the non-insect arthropod orders Opiliones and Chilopoda. Spiders found in the pitfall traps that were not Salticidae or Thomisidae were also classified as epigeal. Foliar predators consisted of all other groups, as well as spiders that were captured in sweep net transects. Individual taxa were also assessed to determine how floral plantings impacted specific groups. 

### 2.5. Data Analysis

Data were analyzed using generalized linear mixed effects models (GLMM) with the glmmADMB package [43] in R (Version 3.3.2) [44]. To determine how floral plantings impact pollinator abundance, separate models were created with overall pollinator abundance in margins, overall pollinator abundance in crops, and bee abundance in margins and in crops as response variables. Abundances of individual pollinator taxa were also used as response variables in separate analyses. Treatment and percent floral cover were used as fixed effects in separate models, where “Treatment” referred to whether field margins were planted with flowers (Flowers), or left unmanaged (Control). For analyses of in-crop pollinators, sampling location was also added as a fixed effect, alternatively as a continuous variable, and as a factor in separate analyses to determine overall effect of distance into the crop and differences between specific sampling locations respectively. Means separation between in-crop locations was achieved by running multiple GLMM comparing only two locations at a time to determine significant differences between each possible location pairing, and applying a Bonferroni correction. Field identity was included as a random effect to account for among-field climatic, soil, and surrounding landscape variability. Sampling date (month nested within year) was also included as a random effects in all models to account for repeated sampling throughout the year respectively. Due to overdispersion of count data, a negative binomial was selected as the distribution, with a log link function. For pollinator taxa where fewer than 10 individuals were collected, no analyses were conducted.

Models for predators were constructed in a similar manner. To determine how floral plantings impact predator abundance, separate models were created with overall predator abundance in margins, overall predator abundance in crops, and foliar and epigeal predator abundances in margins and in crops as response variables. Treatment and percent floral cover were used as fixed effects in separate analyses, and location within crop (alternatively as a continuous variable and as a factor) was added as a fixed effect for in-crop analyses. Field identity and date were used as random effects, and a negative binomial distribution was used for all models. For predator taxa with fewer than 10 individuals, no analyses were conducted. 

## 3. Results

### 3.1. Pollinators

A total of 1819 pollinators were collected from 2017 to 2019. Within field margins, floral plantings resulted in significantly greater pollinator abundance (z = 5, *p* = 5.6 × 10^−7^) (Figure 2a), and higher floral cover corresponded to higher pollinator numbers (z = 4.35, *p* = 1.4 × 10^−5^). Individual taxa responded differently to the presence of flowers in field margins. In all cases, floral plantings either significantly increased abundance within each taxon, or there was no significant effect of treatment (Table 1). Bee abundance was significantly increased by floral plantings (z = 3.81, *p* = 1.4 × 10^−4^) and by higher floral cover (z = 4.51, *p* = 6.6 × 10^−6^). At the family level, higher numbers of Andrenidae were correlated with higher floral cover, while treatment did not have an effect. Apidae abundance increased with higher floral cover, and treatment. Floral plantings led to significantly higher numbers of Syrphidae, and Megachilidae, while floral cover did not have an effect (Table 1). 

Within potato crops, the presence of floral plantings in adjacent margins had no significant effect on pollinator abundance in potato fields (z = −0.72, *p* = 0.47). However, increasing floral cover within field margins led to a marginally significantly greater abundance of pollinators in adjacent potato crops (z = 1.85, *p* = 0.065) (Figure 3). In-crop bee abundance was significantly negatively related to the presence of floral plantings in margins (z = −2.72, *p* = 0.0066). However, there was no effect of floral cover on in-crop bee abundance (z = 0.3, *p* = 0.77). Most individual pollinator taxa had too few individuals for in-crop analyses, although there were significantly more Halictidae in potato crops adjacent to control margins (z = −2.52, *p* = 0.012) and marginally more Bombus (z = −1.68, *p* = 0.092). Increasing distance into potato crops resulted in marginally fewer pollinators (z = −1.77, *p* = 0.076) and bees (z = −1.67, *p* = 0.094) (Figure 2b,c).

### 3.2. Predators

A total of 17,139 predators were collected during this study. Floral plantings in field margins significantly increased overall predator abundance (z = 2.65, *p* = 0.008) (Figure 4a) and foliar predator abundance (z = 2.09, *p* = 0.036) within margins. The amount of floral cover in field margins did not have a significant impact on the abundance of predators overall (z = 0.85, *p* = 0.4), or on foliar predators (z = 1.12, *p* = 0.26). Epigeal predator abundance was significantly increased by the presence of floral plantings (z = 3.76, *p*= 1.7 × 10^−4^), and greater floral cover had no significantly effect on epigeal abundance (z = −0.19, *p* = 0.85). 

The majority of measured taxa either had significantly higher abundance in floral margins, or were unaffected by treatment. At the level of order, Hemiptera and Opiliones had significantly higher abundances in floral margins. Control margins had significantly higher abundances of Coccinellidae (Table 2). Individual taxa had different responses to floral cover. At the level of order, increased floral cover in the margins led to increased abundance of predatory Coleoptera and Hemiptera. Araneae abundance declined with increasing floral cover. The abundance of several families of predators increased with increasing floral cover, such as Cantharidae, Anthocoridae, and Chrysopidae adults. Staphylinidae also increased with higher floral cover, although the effect was only marginally significant (Table 2). 

Within adjacent potato crops, there was no significant effect of treatment on the overall abundance of predators (z = 0.1, *p* = 0.92) (Figure 4b), or on the abundance of foliar (z = −0.22, *p* = 0.82) and epigeal predators (z = 0.54 *p* = 0.59) (Figure 5a,b). Floral cover also had no significant effect on overall predator abundance (z = −1.14, *p* = 0.25), foliar predator abundance (z = −0.36, *p* = 0.72), or epigeal predator abundance (z = −0.9, *p* = 0.37). Individual taxa were predominantly unaffected by treatment, with only Opiliones abundance significantly increased by floral margins (Table 3). Salticidae, Cantharidae, and Staphylinidae abundance increased with increasing floral cover in margins, while Thomisidae abundance declined (Table 3). 

## 4. Discussion

### 4.1. Pollinators and Predators in Margins

We found that both pollinator and predator abundance in field margins were increased by the presence of floral plantings. Our results are similar to numerous previous studies [21,33,45], and may be due to several factors. The seed mixture in this study has a high proportion of native plants and high species richness, which are linked to increased arthropod abundance and diversity in other studies [46,47,48]. This could explain in part why floral plantings here were successful in increasing predator abundance, when they have not always been in other studies [20,49]. Additionally, floral resources such as pollen and nectar are frequently a limiting resource for pollinators in agroecosystems [50,51] and establishing floral plantings can provide the sources of food pollinators require and increase their abundance. 

The effects of floral cover on beneficial arthropod abundance in field margins are more complicated. While increased floral cover resulted in increased pollinator abundance, there was no effect of floral cover on overall predator abundance or foliar or epigeal predator abundance. From our results, it appears that pollinator abundance increases both with the presence of floral plantings and increased floral cover, while predator abundance increases with the presence of floral plantings, and is unaffected by increased floral cover. In previous studies on pollinators, floral plantings that increased floral cover compared to control margins were more attractive to pollinators [24] and floral density can be a stronger predictor of bee abundance than the presence or absence of plantings [25]. From these studies, it appears that a greater number of flowers providing nectar and pollen results in greater bee abundance. For predators, improved microclimate [19], alternative prey [22], or other factors provided by floral plantings may be more important than a greater number of flowers overall (although see [21]). Studies on beetle banks have shown that increased habitat and shelter provided by plantings without flowers can have a positive impact on predators in agricultural settings [52,53]. While not directly measured, the floral margins in our study system were composed of observably more dense, tall, and leafy foliage than control margins in most cases. This increased shelter and physical structure may provide the correct conditions to favor predators more so than increased floral cover. 

The abundances of many individual pollinator taxa were positively correlated with floral margins. For pollinators, the genus *Bombus* increased most significantly in the presence of flowers, similar to the results of many other studies [17,26,54]. Other genera more common in floral margins, such as *Apis*, *Melissodes*, and *Megachile,* are predominantly larger-bodied bees. In general, larger bees are capable of longer flights and greater foraging distances [55] and therefore may be able to seek out patches of resources in the landscape to a greater degree than smaller and less mobile bees can. Floral plantings established in commercial agroecosystems similar to our study system may be more likely to support larger bees that are capable of traveling further. 

Many predatory taxa responded positively to floral plantings as well. Notably, the subfamily Phymatinae showed the greatest increase in relative abundance in floral margins. Phymatinae are ambush predators, and specifically wait on flowers for their prey [56]. Similarly, Anthocoridae has been positively associated both with flowers and greater plant diversity [57,58] and was increased by floral margins and greater floral cover in our study. Two other taxa, Thomisidae and Cantharidae, were also increased in floral margins, and can be linked to a preference for flowers [59,60]. However, both Phymatinae and Thomisidae did not benefit from increasing floral cover as would be expected. Specific flower species found within floral margins may be more favorable towards these taxa, and influence their abundance more than overall floral cover. 

Other predators were more abundant in floral margins that do not have a clear biological relationship to flowers, including several epigeal taxa and epigeal predators overall. For some taxa such as Carabidae, increased vegetation cover in wildflower plantings has led to increased abundance and richness in previous studies [61]. The observed increase in epigeal abundance provides additional evidence that factors other than the presence of flowers per se have a stronger influence on predator abundance within floral plantings. 

Although most predator groups were increased by the presence of floral plantings, Coccinellidae was significantly more abundant in control margins. This is particularly surprising seeing as many ladybeetle species consume pollen and nectar [62,63]. Prey species favoring coccinellids may have been more prevalent in control margins, or microclimatic conditions within grassy areas could have been preferable over conditions within floral plantings. 

### 4.2. Pollinators and Predators in Crops

Floral margins did not lead to greater numbers of pollinators or predators in adjacent crops, unlike the results of previous studies [25,28,33]. For pollinators, this may be at least partially due to the crop type: potato flowers offer no nectar and require sonication to extract pollen, possibly limiting this resource to specific taxa such as bumblebees [64]. Potatoes next to control margins did attract a marginally significantly greater number of *Bombus*, and since control margins were more frequently lacking in floral resources, it follows that bumblebees may forage within adjacent potato fields to a greater extent. For most pollinators however, there may be little incentive to move into adjacent potato fields.

Our results showing more bees in floral margins, but fewer bees within potato crops near floral margins (Figure 6), could also indicate that floral plantings are acting to concentrate bees more than being a source [65]. If floral plantings provide all the necessary resources for bees, and nearby crop fields offer little, it is unlikely that bees will spill over into surrounding crops. Considering many bees will favor and return to resource-rich patches [66], floral plantings may serve to draw bees away from crops. While not necessarily a concern for conservation purposes, floral margins concentrating bees within margins could be problematic if floral plantings are expected to promote pollination services. Related to this point, although floral plantings did increase pollinator and predator abundance in field margins, our study did not determine that floral plantings actually increased populations of pollinators or predators, instead of simply concentrating them in floral margins. Further study about the efficacy of floral plantings to truly increase beneficial insect populations in commercial, conventionally managed agroecosystems would be an important next step based on our work. 

For predators, floral margins had little effect on abundance in potato fields, and almost no taxa measured within crops were significantly affected by this treatment. The one exception was Opiliones, possibly because some are highly mobile predators and spilled over from floral margins [67]. Our results suggest that few predators are moving from crop margins into adjacent fields, contrary to the findings of other studies [68]. Similar to pollinators, this may be due to a lack of resources within potato fields compared to field margins. Additionally, while a full analysis of predator communities is beyond the scope of this study, the predator communities within crops and within margins appeared very different from one another (Figure 7). These differences further suggest that predators are not simply spilling over from margins. Instead, our results indicate that different predator communities inhabit crop fields compared to margins, and that while some spillover likely occurs (as evidenced by higher abundances of predators in at the edge of the crop), the overall communities appear distinct. 

Based on our results, growers and conservationists should evaluate how best to utilize floral plantings to achieve their desired outcomes. From a conservation standpoint, commercially available seed mixtures such as the one used in our study may act to increase both pollinator and predator abundance when established on a large scale in conventionally managed agroecosystems. Additionally, greater floral cover provided by floral plantings will likely result in greater pollinator abundance, and should be encouraged if the goal is related to pollinator conservation. However, if the intention is to promote predators, the presence of plantings appears to be more impactful than increased floral cover, and the necessity of flower-dense plantings is more questionable. Finally, while such floral plantings increase beneficial inset abundance within margins, there were limited effects in nearby potato crops, and the number of bees in the crop actually decreased. Growers seeking to increase pollination or biological control services may need to consider how best to encourage beneficial insects to spill over into adjacent crops instead of simply creating floral plantings. 

## 5. Conclusions

Floral plantings can promote pollinators and predators within commercial agroecosystems, while limiting disruption to growers. Our results show that when floral plantings are established on a large scale on commercial farms, they increase the overall abundance of both pollinators and predators. In particular, pollinators respond positively to increased floral cover, whereas predators appear to benefit from other factors associated with floral plantings. The numbers of beneficial insects did not increase in nearby potato fields however, indicating that floral plantings may act to concentrate pollinators and predators rather than export them to surrounding areas. These results can help policy makers and growers determine how best to manage land in order to provide both for the conservation of beneficial insects and for the ecosystem services that they provide in commercial agroecosystems.

## Figures and Tables

**Figure 1 insects-12-00091-f001:**
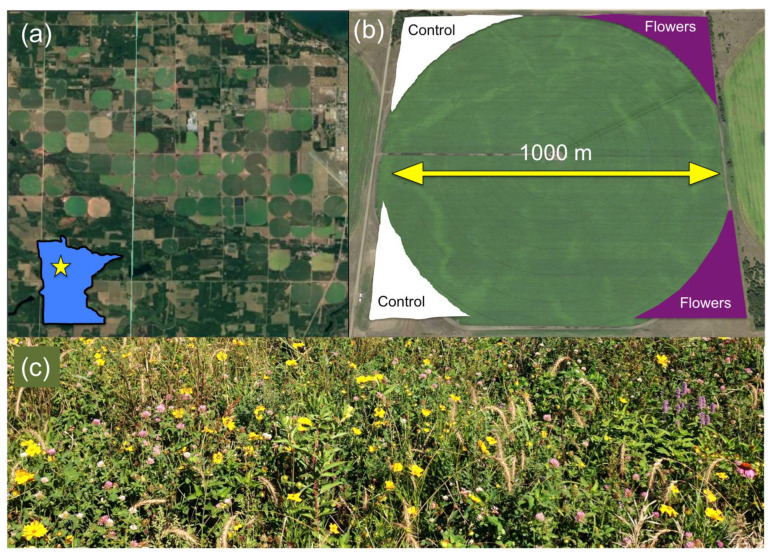
(**a**) Aerial image of representative section of study agroecosystem, and location within Minnesota designated by star. (**b**) Typical configuration of floral (purple) and control margins (white) around fields. (**c**) Typical floral margin mid-season.

**Figure 2 insects-12-00091-f002:**
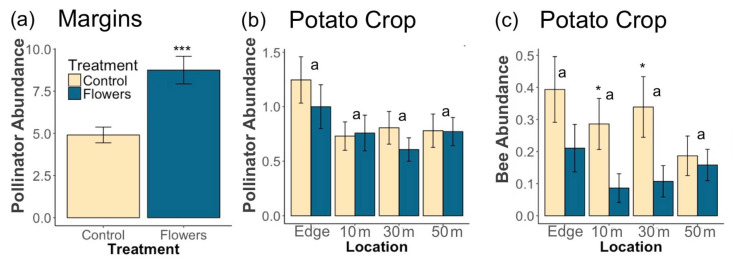
Mean (±1 SE) abundance of (**a**) pollinators in field margins, (**b**) pollinators within potato crops, and (**c**) bees within potato crops. Asterisks denote significant differences between treatments at each location (* *p* < 0.05, *** *p* < 0.001). Letters denote significant differences in overall abundance between locations.

**Figure 3 insects-12-00091-f003:**
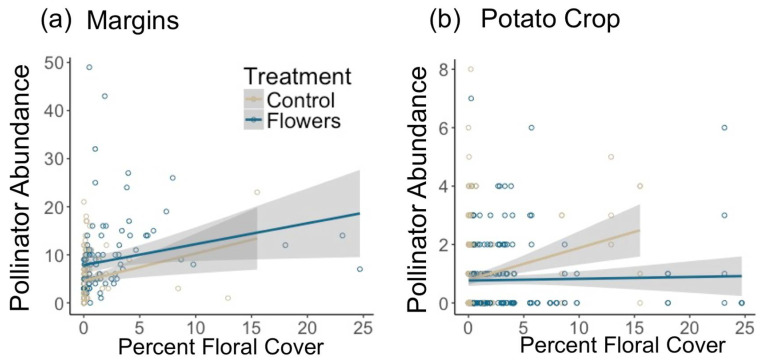
Effect of floral cover on pollinator abundance in (**a**) field margins and (**b**) potato crops. Shaded area represents 95% confidence interval.

**Figure 4 insects-12-00091-f004:**
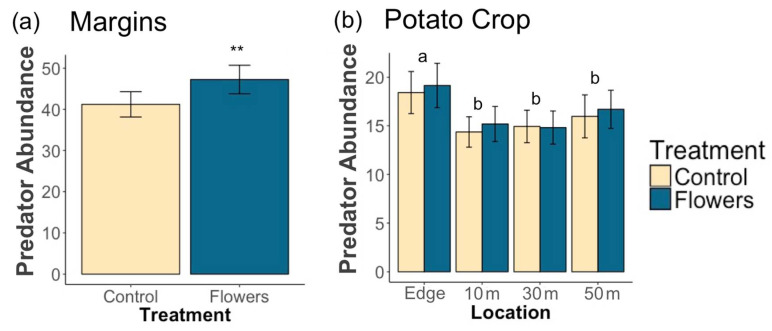
Mean (±1 SE) abundance of (**a**) predators in field margins and (**b**) predators within potato crops. Asterisks denote significant differences between treatments at each location (** *p* < 0.01). Letters denote significant differences in overall abundance between locations.

**Figure 5 insects-12-00091-f005:**
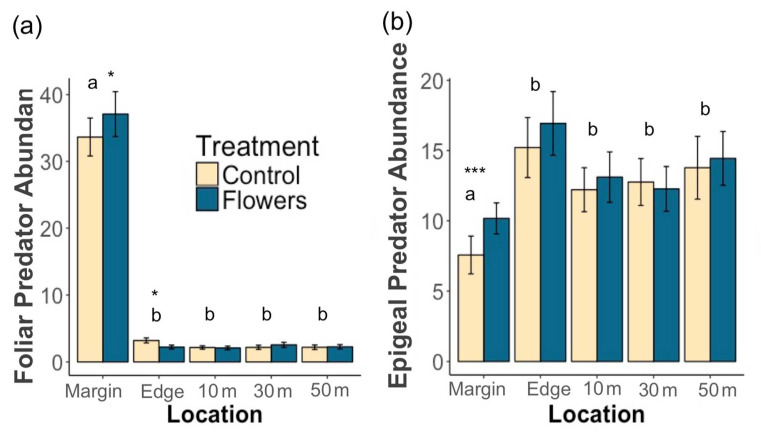
Mean (±1 SE) (**a**) epigeal predator abundance, and (**b**) foliar predator abundance. Asterisks denote significant differences between treatments at each location (* *p* < 0.05, *** *p* < 0.001). Letters denote significant differences in overall abundance between locations.

**Figure 6 insects-12-00091-f006:**
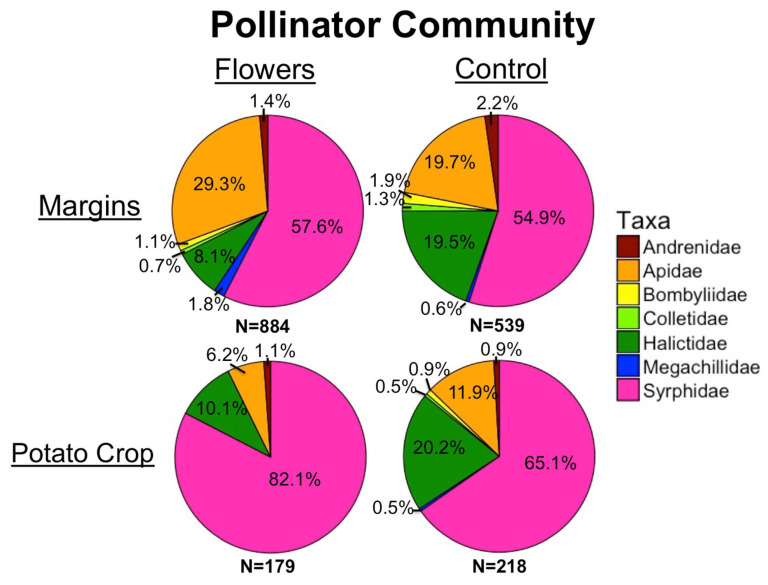
Pollinator community in margins and potato crops, with and without floral plantings.

**Figure 7 insects-12-00091-f007:**
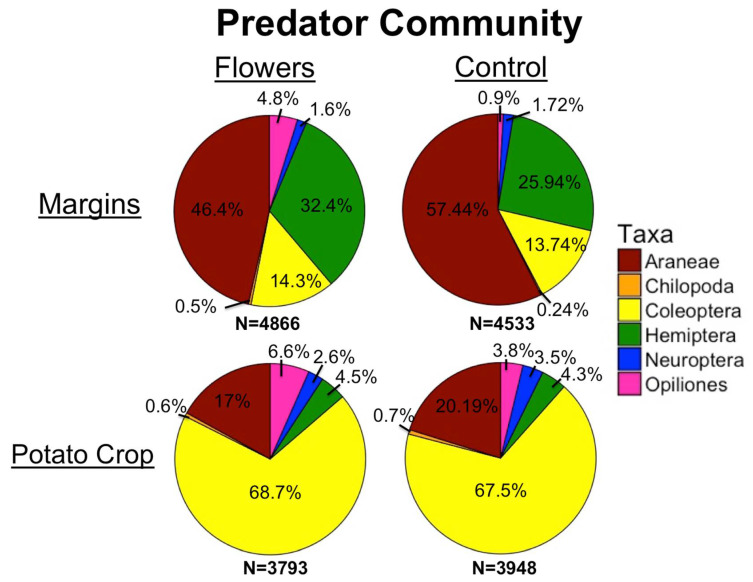
Predator community in margins and potato crops, with and without floral plantings.

**Table 1 insects-12-00091-t001:** Pollinator abundance, z values, and significance by treatment and floral cover within field margins. Bolded values are significant (*p* < 0.05). Taxa with fewer than 10 individuals were not analyzed.

		Margins					
Pollinators		Numbers of Individuals		Treatment		Floral Cover	
Family	Genus	Flowers	Control	z Value	Significance	z Value	Significance
Bombyliidae		10	10	0.67	0.5	−0.16	0.88
Syrphidae		509	296	4.06	**4.9 × 10^−5^**	1.38	0.16897
Andrenidae		12	12	−0.22	0.82	3.53	**4.1 × 10^−4^**
	*Andrena*	12	10	0.13	0.9	2.73	**0.0063**
	*Calliopsis*	0	1	---	---	---	---
	*Perdita*	0	1	---	---	---	---
Apidae		259	106	5.15	**2.60 × 10^−7^**	4	**6.40 × 10^−5^**
	*Anthophora*	0	1	---	---	---	---
	*Apis*	88	33	2.43	**0.015**	3.28	**0.001**
	*Bombus*	128	49	4.17	**3.1 × 10^−5^**	3.78	**1.5 × 10^−4^**
	*Ceratina*	5	12	−0.99	0.32261	−0.93	0.3535
	*Epeolus*	1	0	---	---	---	---
	*Melissodes*	36	10	2.86	**0.0042**	2.69	**0.0071**
	*Nomada*	1	1	---	---	---	---
Colletidae		6	7	−0.22	0.82	0.82	0.41
	*Colletes*	0	1	---	---	---	---
	*Hylaeus*	6	6	0.02	0.98	−1.24	0.22
Halictidae		72	105	−0.5	0.62	0.01	0.988
	*Agapostemon*	6	4	0.71	0.47	−1.06	0.29
	*Augochlora*	1	0	---	---	---	---
	*Augochlorella*	7	11	−0.29	0.77138	−1.14	0.25394
	*Halictus*	4	7	−0.44	0.66	0.25	0.81
	*Lassioglossum*	52	79	−1.03	0.303	−0.47	0.6349
	*Sphecodes*	2	4	−0.51	0.613	2.02	**0.0429**
Megachilidae		16	3	2.69	**0.0071**	1	0.32
	*Ashmeadiella*	1	0	---	---	---	---
	*Heriades*	0	1	---	---	---	---
	*Hoplitis*	2	0	---	---	---	---
	*Megachile*	10	1	2.39	**0.017**	1	0.3164
	*Osmia*	3	1	---	---	---	---

**Table 2 insects-12-00091-t002:** Predator abundance, z values, and significance by treatment and floral cover within field margins. Bolded values are significant (*p* < 0.05). Taxa with fewer than 10 individuals were not analyzed.

			Margins					
Predator			Number of Individuals		Treatment		Floral Cover	
Order	Family	Genus	Flowers	Control	z Value	Significance	z Value	Significance
Araneae			2260	2604	−0.39	0.7	−2.69	**0.0072**
	Salticidae		156	212	−0.11	0.91	0.65	0.52
	Thomisidae		271	180	3.22	**0.0013**	0.39	0.696
	Other spp		1833	2212	−0.91	0.36	−3.34	**8.5 × 10^−4^**
Chilopoda			22	11	1.76	0.079	0.68	0.5
Coleoptera			697	623	1.6	0.11	2.85	**0.0043**
	Cantharidae		209	100	1.96	0.0501	2.39	**0.017**
	Carabidae		218	132	3.68	**2.3 × 10^−4^**	−0.08	0.94
	Coccinellidae		96	289	−3.39	**6.9 × 10^−4^**	1.29	0.2
		Adults	79	169	−2.68	**0.0074**	1.58	0.11
		Larvae	17	120	−2.39	**0.017**	−0.12	0.903
	Staphylinidae		174	102	2.51	**0.012**	1.72	0.085
Hemiptera			1579	1176	3.62	**3.00 × 10^−4^**	2.97	**0.003**
	Anthocoridae		759	680	2.1	**0.036**	3.83	**1.3 × 10^−4^**
	Lygaeidae	*Geocoris* spp	77	80	0.88	0.377	−0.9	0.366
	Nabidae		255	232	1.29	0.2	−0.28	0.78
	Pentatomidae	*Podisus maculiventris*	1	0	---	---	---	---
	Phymatinae		245	97	4.88	**1.00 × 10^−6^**	1.05	0.29
	Reduviidae		242	87	5.5	**3.70 × 10^−8^**	0.58	0.56
Neuroptera			76	78	0.26	0.798	1.4	0.1617
	Chrysopidae		73	77	0.09	0.926	1.38	0.1668
		*Chrysopa* spp Adult	25	27	−0.24	0.81	2.11	0.035
		*Chrysopa* spp Larvae	48	50	0.24	0.8137	0.7	0.48242
	Hemerobiidae		3	1	---	---	---	---
Opiliones			232	41	4.71	**2.50 × 10^−6^**	0.47	0.642

**Table 3 insects-12-00091-t003:** Predator abundance, z values, and significance by treatment and floral cover within potato crops. Bolded values are significant (*p* < 0.05). Taxa with fewer than 10 individuals were not analyzed.

			Potato Crop					
Predator			Number of Individuals		Treatment		Floral Cover	
Order	Family	Genus	Flowers	Control	z Value	Significance	z Value	Significance
Araneae			644	797	−0.19	0.84577	−1.11	0.26908
	Salticidae		25	24	0.09	0.93	2	**0.045**
	Thomisidae		54	62	0.09	0.93	−2.2	**0.028**
	Other spp		565	711	−0.26	0.7913	−0.75	0.454
Chilopoda			23	29	−0.17	0.86	0.74	0.46
Coleoptera			2606	2664	0.04	0.97	−0.68	0.5
	Cantharidae		7	9	0.12	0.902	2.01	**0.045**
	Carabidae		1493	1590	−0.03	0.97	0.35	0.72
	Coccinellidae		26	30	−0.06	0.95	−0.06	0.96
		Adults	25	24	0.53	0.6	0.21	0.83
		Larvae	1	6	−1.46	0.14497	−0.59	0.55305
	Staphylinidae		1080	1035	0.85	0.3972	−2.01	**0.04473**
Hemiptera			169	169	−0.14	0.892	−1	0.315
	Anthocoridae		153	147	0.36	0.716	−1.14	0.2528
	Lygaeidae	*Geocoris* spp	3	3	−0.26	0.8	−0.01	0.99
	Nabidae		9	14	−0.7	0.49	0.12	0.9
	Pentatomidae	*Podisus maculiventris*	0	0	---	---	---	---
	Phymatinae		0	3	---	---	---	---
	Reduviidae		4	2	0.89	0.37	−0.17	0.87
Neuroptera			99	139	−0.82	0.4145	0.64	0.5223
	Chrysopidae		96	135	−0.85	0.398	0.57	0.5694
		*Chrysopa* spp Adult	33	42	−0.46	0.65	0.41	0.68
		*Chrysopa* spp Larvae	63	93	−1.12	0.2635	0.08	0.9331
	Hemerobiidae		3	4	−0.31	0.75	0.9	0.367
Opiliones			252	149	4.05	**5.2 × 10^−5^**	0.8	0.423

## Data Availability

The data presented in this study are available on request from the corresponding author. The data have not yet been uploaded to an online repository, and are therefore not yet publicly available.

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
