# Peer review of "Floral Plantings in Large-Scale Commercial Agroecosystems Support Both Pollinators and Arthropod Predators"

_insects, 2021, doi:10.3390/insects12020091_

Round 1

Reviewer 1 Report

This is an interesting and well conducted study.

However, I there are some issues that should be addressed before it can be accepted for publication-

MAJOR CONCERNS

1) For predators the authors report results for margins (Table 3) and potato crops (Table 4), whereas for pollinators only results for margins are given (table 2). Why?

In the discussion they refer to the abundance of Bombus in potato crops (line 353) but there is no analysis for this. I assume the authors missed to include a Table with results for pollinators in potato crops.

2) Figures 6 and 7 should be placed in the Results section. Also, I suggest to better develop these results. For example, the authors might perform chi-square tests with contingency tables (or other appropriate analyses) to show that proportions of taxonomic composition vary.

3) authors conclude that “floral plantings may act to concentrate pollinators and predators rather than export them to surrounding areas” and that “these results can help policy-makers and growers determine how best to manage lands”, but do not explain how. I think this part should be expanded in the Discussion. I suggest adding a final paragraph in the Discussion to explain the practical consequence of the results of this study.

MINOR CONCERNS

There are many points that require author attention to improve readability:

56, 71: “number”: do you mean species richness or abundance? (or both?)

79: “can provide for pollinators” Please revise

85-87: Can you provide some reference to support that there is little research on this?

90-91: Can you provide some reference to support that there  are few studies?

134-136: This point is not very clear. Did you place ONE transect in each corner? That is, four aligned sampling points in each corner? Please, clarify. If possible, please add examples of location of sampling points on Figure 1 b. As the scale may be a problem, you can specify that the diagrammatic representation is not to scale.

140: For inflorescence, did you use the size of individual flowers and then multiplied this for the average number of flowers per inflorescence?

144: Can you indicate where the starting points were located? Were those close to the corners or in a point of the circumference between the two corners? Again, please add examples of location of sampling points on Figure 1 b.

144: It is not clear. Did you use 5 transects, i.e. one transect in the margin, one at 0 m, one at 10 m, one at 30 m and one at 50 m? Or were these collecting points along a single transect? Also, what do you mean with “margins”? I assume the corners. Also, I assume that you sampled all four corners, right?

146: How long was each pollinator transect? How many transects per crop you used?

148: Why timing differed between transects in the margins (5 minutes) and transects within the potato crop (3 minutes)

150: How many sweep-net transects you used per crop?.

226-230: This sentence is obscure to me. Do you mean that the presence of flowers did not affect pollinators in the margins (Figure 3 a), whereas it affected positively pollinators within the crops (Figure 3 b)?

Table 1: Caption: “via backwards elimination and comparison of AIC values”. I think you mean “via backwards elimination based on AIC values”. Correct?

Please, explain what is Est. Std. Also, for Est. Std. and p always use the same number of digits (I suggest three are enough)

Table 2 You used here the expression “z values” and “significance”, whereas in table 1 you used z and p. Please, be consistent.

Names of genera should be in italics.

Tables 3-4. See comments to Table 2

Figure 2. Please, correct the left margin of panels (a) and (c). Some problems occurred when you imported/saved the figure (there is vertical blue bar and a dashed line). Use of asterisks and letters to denote significance is not very clear. Based on letters, in potato crops, for pollinator abundance (panel b), it seems that the edge had significantly higher values than all other locations, which, by contrast, were similar to each other; at the same time, there was no difference between treatment and control within the edge. For bee abundance (panel c), it appears that there is a difference between edge and 50 m, whereas 10 m and 30 m did not differ from either edge or 50 m. However, in 10 m and 30 m there were differences between control and treatment, whereas there were no differences between control and treatment in edge and 50 m. Is this interpretation correct? If so, I suggest reporting it in figure caption, to help the reader to understand the graphs.

Figure 4. See comments to Figure 2 to improve readability.

Figure 5. Some problems occurred with the y axis (also on the right side of panel b). See also comments to Figure 2 to improve readability.

Figure 3. Some problems occurred with the y axis.

LANGUAGE/TYPO/GRAPHIC CORRECTIONS

Line 55: lead to both a loss of insect abundance and of ecosystem function [14]. -> lead to a loss of both insect abundance and of ecosystem functions [14].

110: Figure. 1 -> Figure 1

208: Figure. 2A -> Figure 2 a

215: Megachillidae -> Megachilidae [one el]

234: Figure. 2B, C -> Figure 2 b, c

248: Figure. 4A -> Figure 4 a

250-251, 275-276: Use three digits for p

267: Chysopidae -> Chrysopidae

275: Figure. 4B -> Figure 4 b

276: Figure. 5A, B -> Figure 5 a, b

358: Figure. 6 -> Figure 6

374: Figure. 7 -> Figure 7

Figure 1 c has some black borders and the left side is not perfectly aligned. Please, correct.

Figure 6: Correct N=884 and N=539

Appendix A. Use always two digits. E.g. 4.3 -> 4.30

Author Response

Responses to the reviewer's comments are below.
The reviewer's comments are Bolded and Underlined, while the responses are in plain text.

Reviewer 1
MAJOR CONCERNS

1) For predators the authors report results for margins (Table 3) and potato crops (Table 4), whereas for pollinators only results for margins are given (table 2). Why?
As stated on line 232-233, the majority of pollinator taxa had too few individuals to conduct analyses within potato crops, and for taxa where there were enough individuals, the pertinent results were covered in the text. More predatory taxa and more individuals in each predator taxa were found in potato crops, warranting a table showing the results. Because of this, the authors chose to not to include the table for in-crop pollinators as it was mostly blank.

In the discussion they refer to the abundance of Bombus in potato crops (line 353) but there is no analysis for this. I assume the authors missed to include a Table with results for pollinators in potato crops.
Added in the missing data on Bombus to the results section “and marginally more Bombus (z=-1.68, p=0.092) [in crops adjacent to control margins]”.

2) Figures 6 and 7 should be placed in the Results section. Also, I suggest to better develop these results. For example, the authors might perform chi-square tests with contingency tables (or other appropriate analyses) to show that proportions of taxonomic composition vary.
Figures 6 and 7 were left in the discussion section as they are just modifications of data presented in the tables in the results section. The authors think a full community is beyond the scope of this current study. Added “While a full analysis of predator communities is beyond the scope of this study,”

3) authors conclude that “floral plantings may act to concentrate pollinators and predators rather than export them to surrounding areas” and that “these results can help policy-makers and growers determine how best to manage lands”, but do not explain how. I think this part should be expanded in the Discussion. I suggest adding a final paragraph in the Discussion to explain the practical consequence of the results of this study.
Added “Based on our results, growers and conservationists should evaluate how best to utilize floral plantings to achieve their desired outcomes. From a conservation standpoint, commercially available seed mixtures like the one used in our study may act to increase both pollinator and predator abundance when established on a large scale in conventionally managed agroecosystems. Additionally, greater floral cover provided by floral plantings will likely result in greater pollinator abundance, and should be encouraged if the goal is related to pollinator conservation. However, if the intention is to promote predators, the presence of plantings appears to be more impactful than increased floral cover, and the necessity of flower-dense plantings is more questionable. Finally, while such floral plantings increase beneficial inset abundance within margins, there were limited effects in nearby potato crops. Growers seeking to increase pollination or biological control services may need to consider how best to encourage beneficial insects to spill over into adjacent crops instead of simply creating floral plantings.”

MINOR CONCERNS

There are many points that require author attention to improve readability:

56, 71: “number”: do you mean species richness or abundance? (or both?)
Changed to “species richness” on line 56, and to “abundance” on line 71

79: “can provide for pollinators” Please revise
Changed to “provide resources for”

85-87: Can you provide some reference to support that there is little research on this?
This statement was made based on a review of the literature done by the authors, and not based on an existing published review or other paper stating little research has been done.

90-91: Can you provide some reference to support that there are few studies?
Similar to above, this statement was made based on a review of the literature done by the authors, and not based on an existing published review or other.

134-136: This point is not very clear. Did you place ONE transect in each corner? That is, four aligned sampling points in each corner? Please, clarify. If possible, please add examples of location of sampling points on Figure 1 b. As the scale may be a problem, you can specify that the diagrammatic representation is not to scale.
In each corner, 1 transect consisting of 4 aligned sampling points was used to determine floral cover. Added “Starting in the center of each corner margin,” to the beginning of the sentence to clarify this. Did not add visuals of the sampling points to figure 1B as they would be very small (even if not to scale) and would likely be more confusing.

140: For inflorescence, did you use the size of individual flowers and then multiplied this for the average number of flowers per inflorescence?
The average size of the inflorescence was used. To clarify this point, changed the sentence to “Floral cover was measured by counting the total number of flowers or inflorescences (depending on species) and calculating the average flower or inflorescence area for each species using flower size data available at Minnesota Wildflowers, verified with in-field observation.”

144: Can you indicate where the starting points were located? Were those close to the corners or in a point of the circumference between the two corners? Again, please add examples of location of sampling points on Figure 1 b.
Added “Starting in the center of each corner margin, and moving towards the crop” to clarify where the floral transect took place. For the same reason listed above, visuals of the sampling points were not added to figure 1B.

144: It is not clear. Did you use 5 transects, i.e. one transect in the margin, one at 0 m, one at 10 m, one at 30 m and one at 50 m? Or were these collecting points along a single transect? Also, what do you mean with “margins”? I assume the corners. Also, I assume that you sampled all four corners, right?
Added “At each location” to clarify that separate pollinator and sweep net transects were conducted in the margins, and at the edge, 10, 30 and 50 meters into the crop. The phrase “margins” is used in the text beforehand and indicated in figure 1B to refer to the corners of cropping fields. To additionally clarify this, “Starting in the center of each corner margin” was added when discussing floral transects.

146: How long was each pollinator transect? How many transects per crop you used?
Each pollinator transect was based on time, not distance traveled. As stated below, transects were 5 minutes in margins and 3 minutes in the potato crop. Within the crop, 4 transects took place (edge, 10, 30 and 50 meters).

148: Why timing differed between transects in the margins (5 minutes) and transects within the potato crop (3 minutes)
The different timing was based on experience from initial sampling trips. The authors found that 5 minute pollinator transects inside potato crops rarely caught any pollinators, and therefore shortened the amount of time spent on pollinator transects in the crop. Additionally, shorter transects made sampling much more efficient, and allowed the authors to stay on schedule. These additional details were excluded from the text for brevity.

150: How many sweep-net transects you used per crop?
Like the pollinator transects, 4 transects took place within the crop (edge, 10, 30 and 50 meters).

226-230: This sentence is obscure to me. Do you mean that the presence of flowers did not affect pollinators in the margins (Figure 3 a), whereas it affected positively pollinators within the crops (Figure 3 b)?
This sentence is meant to show that the presence of floral plantings did not increase pollinator abundance within crops, but increased floral cover did increase pollinator abundance within crops. The use of “on their own” is to illustrate that although floral plantings as a factor had no significant effect, increased floral cover did have a significant effect on in-crop pollinator abundance.

Table 1: Caption: “via backwards elimination and comparison of AIC values”. I think you mean “via backwards elimination based on AIC values”. Correct?
Per changes suggested by another reviewer, ended up removing Table 1

Please, explain what is Est. Std. Also, for Est. Std. and p always use the same number of digits (I suggest three are enough)
Per changes suggested by another reviewer, ended up removing Table 1

Table 2 You used here the expression “z values” and “significance”, whereas in table 1 you used z and p. Please, be consistent.
Standardized to “significance”

Names of genera should be in italics.
Italicized genera names in the pollinator table

Tables 3-4. See comments to Table 2
Standardized to “significance”

Figure 2. Please, correct the left margin of panels (a) and (c). Some problems occurred when you imported/saved the figure (there is vertical blue bar and a dashed line). Use of asterisks and letters to denote significance is not very clear. Based on letters, in potato crops, for pollinator abundance (panel b), it seems that the edge had significantly higher values than all other locations, which, by contrast, were similar to each other; at the same time, there was no difference between treatment and control within the edge. For bee abundance (panel c), it appears that there is a difference between edge and 50 m, whereas 10 m and 30 m did not differ from either edge or 50 m. However, in 10 m and 30 m there were differences between control and treatment, whereas there were no differences between control and treatment in edge and 50 m. Is this interpretation correct? If so, I suggest reporting it in figure caption, to help the reader to understand the graphs.
Adjusted figure to correct issues. The interpretation of figure 2b and c is correct. Added “Asterisks denote significant differences between treatments at each location” for additional clarity

Figure 4. See comments to Figure 2 to improve readability.
Adjusted figure to correct issues and added “Asterisks denote significant differences between treatments at each location” for additional clarity

Figure 5. Some problems occurred with the y axis (also on the right side of panel b). See also comments to Figure 2 to improve readability.
Adjusted figure to correct issues and added “Asterisks denote significant differences between treatments at each location” for additional clarity

Figure 3. Some problems occurred with the y axis.
Adjusted figure to correct issues

LANGUAGE/TYPO/GRAPHIC CORRECTIONS

Line 55: lead to both a loss of insect abundance and of ecosystem function [14]. -> lead to a loss of both insect abundance and of ecosystem functions [14].
Changed to “lead to a loss of both insect abundance and of ecosystem function”

110: Figure. 1 -> Figure 1
Changed to “Figure 1”

208: Figure. 2A -> Figure 2 a
Changed to “Figure 2a”

215: Megachillidae -> Megachilidae [one el]
Changed to “Megachilidae”

234: Figure. 2B, C -> Figure 2 b, c
Changed to “Figure 2b, c”

248: Figure. 4A -> Figure 4 a
Changed to “Figure 4a”

250-251, 275-276: Use three digits for p
Left p values unaltered as different numbers of digits are used in all tables that report p values and throughout the rest of the paper.

267: Chysopidae -> Chrysopidae
Changed to “Chrysopidae”

275: Figure. 4B -> Figure 4 b
Changed to “Figure 4b”

276: Figure. 5A, B -> Figure 5 a, b
Changed to “Figure 5a, b”

358: Figure. 6 -> Figure 6
Changed to “Figure 6”

374: Figure. 7 -> Figure 7
Changed to “Figure 7”

Figure 1 c has some black borders and the left side is not perfectly aligned. Please, correct.
Adjusted figure to correct issues

Figure 6: Correct N=884 and N=539
Adjusted figure to correct issues

Appendix A. Use always two digits. E.g. 4.3 -> 4.30
Changed appendix to standardize number of decimal places.

Reviewer 2 Report

insects-1084679 review

In the manuscript entitled “Floral plantings in large-scale commercial agroecosystems support both pollinators and arthropod predators,” Middleton et al. investigated whether field margins of potato fields planted with pollinator-oriented flowering seed mixes enhanced the abundance of insect pollinators and arthropod predators, both in the field margins themselves and the surrounding area of the potato field. They found that the margins themselves were, indeed, generally associated with increased abundances of both beneficial arthropod groups, but for the most part, this increased abundance in the field margins did not translate into spillover into the crop field.

As the authors duly pointed out, there exist a body of literature that examines the potential of field-margin flower plantings to enhance beneficial arthropod conservation and ecosystem service delivery on agricultural lands. However, I agree with the authors that more studies on the potential utility of flower plantings in extensively modified, large-scale agricultural landscapes are still needed. The present manuscript thus serves the purpose of filling this data gap. I also found the manuscript to be well written and well organized overall, and the ideas clearly communicated. I do have some issues that I believe need to be adequately addressed before the study could be published. None constitute fatal flaws in my opinion, and most of my more major concerns pertain to clarification of methodological and statistical approaches. My comments are as follows, hopefully they are helpful to the authors as they revise the manuscript.

Major / general comments:

1. I would assume, naively, that treatment (i.e., floral planting vs. control) and % floral cover should be quite highly correlated (see L181). Figure 3 seems to suggest that at least the maximum values of floral cover are much higher in the floral planting margins. However, I don’t see any indication that an analysis was performed to explicitly examine the effect of treatment on % floral cover. I expect that if the floral plantings were at all successful, there should be a multicollinearity issue in including both treatment and % floral cover in the model; however, the authors do not discuss this potential issue at all. Conversely, if multicollinearity is not an issue, it means that floral plantings actually failed to systematically increase % floral cover, which would be interesting as well and would pose a challenge for how the results are currently interpreted by the authors (i.e., with the assumption that floral plantings increased floral resources). I think it is necessary for this issue to be discussed explicitly and carefully.

2. Related to the above, the authors also stated that they collected data on what plant species were blooming during each survey event, as well as the relative abundances of each species, but did not present any analyses of these data. I recognize that the paper is already long and the discussion quite rich without this component, but I do wonder if some of the results could be best explained by flower diversity or the presence of particular taxa (as the authors also speculated), so it would be great if the authors could at least make some use of this strength in their dataset.

3. Also related to the major issue #1: there appear to be both additive and interactive effects of treatment and % floral cover. I think it is very interesting that sometimes these factors act additively and sometimes interactively, and would be very interested in seeing the authors discuss these two distinct effects—of course, assuming that there are no issues with multicollinearity.

4. I don’t see study year listed explicitly as a fixed or random effect; please explain. The authors stated that month nested within year is a random effect, but if I correctly recall how nested random effects work, the author’s reported nesting scheme treats the same month in different years as distinct (i.e., May 2017 is treated as distinct from May 2018 and May 2019), while not explicitly accounting for the influence of year (i.e., the model does not consider the fact that different study years may have different underlying conditions that act across all fields in all sampling dates). Also, if I correctly recall, the nesting scheme disregards any commonality shared across each month in distinct years by treating all months in all years as distinct entities, which I don’t think is the right approach, as one WOULD expect that May in all years should share some characteristics in common, and June across all years share other characteristics, and so on. It seems to me that month and year should be CROSSED random effects rather than NESTED random effects in order to properly partition variance.

5. Tangentially related to #4 above, since the data structure allows, it may even be worth including month as a fixed effect (probably categorical) to examine how any of the effects detected varies with season. I wonder if some of the most interesting patterns may be masked by treating month simply as a random term, instead of taking a closer look at how dynamics may potentially shift month to month. For example, perhaps during certain months, control margins consistently have higher floral cover than floral planting margins, and then vice versa in other months, such that in the aggregate, these patterns are lost. Also it may be interesting and relevant to isolate months during which potato flowers are in bloom to examine pollinator activity within the crop fields. I would encourage the authors to consider how best to make use of the available temporal resolution in their data to make more sense of their findings.

6. The authors state in L123-124, “To minimize among-field climatic, soil, and surrounding landscape variability, floral margins were compared to control margins for the same field.” This sounds like some sort of paired design, and as far as I know, the mixed effects models framework does not allow for constraining comparisons only among field margins in the same field. Please clarify. I would also like to see sample sizes for how many floral planting vs. control field edges there are in each year, and also, if possible, some discussion on whether fields that had more flower margins saw any benefits compared to fields that had only one planted margin.

Minor / specific comments:

This is a well-written and well-constructed manuscript! There are, however, still a few errors scattered here and there (e.g., L15 “plantings” should be “planting”, L211 “taxa” should be “taxon”, L267 “Chysopidae” should be “Chrysopidae”, L399 has a garbled “This research was funded by We acknowledge the support and funding from”; just to name the ones I picked up), so I would encourage the authors to carefully double-check.

L58: I’m not quite sure how to interpret “creating more sustainable existing agroecosystems” since “creating” and “existing” seem to contradict each other in this sentence. Please clarify?

L80-82: The way this sentence is structured seems to suggest that studies have always found that field margin plantings indeed promote both pollinators and predators, and they always spill over into crops. Is that the case? If not the case, please provide counterexamples or caveats?

L86: If not on-farm in conventionally managed landscapes, does that mean that previous work have just been done in experimental field settings? If that’s the case, perhaps that should be made a little clearer in the segment above (see also my comments re: L80-82).

Figure 1a: If possible, please put the map of Minnesota as an inset; perhaps a little smaller so that surrounding states could be shown. The current “floating” map looks somewhat awkward and may be confusing for those not familiar with US geography.

L134-135: Please comment on where this 15 m transect is placed in the margin? I assume that, like any ecotone, there are edge effects.

L135-136: 4 quadrats of 1 square meter seems adequate for this sampling scale if floral cover is very uniform, but perhaps inadequate if floral cover is highly patchy (which is the case in many natural systems). Could you comment on the uniformity of floral cover / species composition? Perhaps simply examining the variation that exists among the 4 quadrats in the same margin would provide a good answer.

L137-138: Verified via website using digital imagery taken in the field? Or collected samples?

L144-145: I’m not quite clear on whether there is one set of within-crop sampling per crop field (i.e., 1 per circular planting of potatoes) or one set per field margin (i.e., 4 per circular planting of potatoes). Please clarify.

L146: Are the “pollinator transects” actually “transects”? It sounds like it’s just walking around collecting pollinators (nothing wrong with that). I generally see “transects” as formally laid out paths that have a start and end point and that are used consistently, which does not seem to apply to this situation (at least, as written). It would be helpful to know if the total distance covered is kept consistent, or only sampling time (e.g., if no pollinators are seen, the observer walks more quickly).

L147: Table 2 shows that there are Bombyliidae flies, which are bee flies, not hover flies.

L150-151: Similar to the above comment, it would be useful to know if the distance walked is kept constant.

L151-152: Why pitfall traps? While pitfall traps have been known to collect pollinators (there are a few published studies out there), pollinator trapping usually involves painted bowl traps. Again, nothing wrong with this approach (plus, the sampling is already done), but it would be good to see a justification / explanation for why the authors took this approach. If the pitfalls were primarily used for predators, but incidentally also captured pollinators, that should be more clearly stated.

L187-189: There are tools (e.g., emmeans package in R) to run these multiple pairwise comparisons; they should be considered. Otherwise, the authors are encouraged to apply some sort of multiple comparison correction themselves.

L199-200: Is location within crop also included alternatively as categorical factor and as continuous variable for predators, like it was for pollinators?

L202-203: I am alas unable to access the supplementary information files. However, I do want to know if model selection via delta-AIC has caused any random effects to be dropped from modeling?

L227-228: Looking at Figure 3b, I would interpret this finding to mean that increasing cover ONLY affects pollinator abundance in crops adjacent to control margins, and NOT in crops adjacent to flower margins. When there’s such a strong interaction, the main effect result probably can’t be trusted. You can verify by checking each treatment level on its own (i.e., separate GLMMs).

Figure 2, Figure 3, Figure 5: There are some graphical issues here; please fix.

L313: What are beetle banks?

L325-326: These are excellent points. The points also touch on the discussion of whether these plantings simply draw in pollinators from the landscape to concentrate in an area of rich flowers (i.e., a behavioral response) or if they actually “support” pollinator populations by actually increasing reproductive output of pollinators drawn to the plantings. It will be helpful for the authors to discuss this point (not necessarily at length, though I think acknowledging this data gap is important). Same logic with L384 where the authors state that floral plantings “increase the overall abundance of both pollinators and predators”—given the spatial scale considered, and the fact that planted and unplanted field margins seem to be intermixed, it seems plausible that the difference in abundance simply reflects movement of individuals from an untreated margin into a floral planting margin, without any overall abundance gains at the landscape scale.

L350-351: This discussion reminds me: were there any significant sources of floral resources in bloom within the potato crop fields throughout all of the survey times? I would imagine that the potato blooming time is brief relative to the whole field season. This seems to point towards separating out the time points of crop bloom and analyzing those time points separately from other points of in-crop pollinator sampling. And if in-crop pollinator sampling was done at a time when there is negligible floral resources available in the crop, this should also be explicitly discussed. In fact, a more complete discussion on what floral resource conditions are like in the crop field overall would be appreciated.

L402-404: Isn’t this funding information more appropriate in the funding section immediately above?

Author Response

Responses to reviewer's comments are below.
Original reviewer comments are Bolded and Underlined, while authors' replies are in regular type.

Reviewer 2
In the manuscript entitled “Floral plantings in large-scale commercial agroecosystems support both pollinators and arthropod predators,” Middleton et al. investigated whether field margins of potato fields planted with pollinator-oriented flowering seed mixes enhanced the abundance of insect pollinators and arthropod predators, both in the field margins themselves and the surrounding area of the potato field. They found that the margins themselves were, indeed, generally associated with increased abundances of both beneficial arthropod groups, but for the most part, this increased abundance in the field margins did not translate into spillover into the crop field.

As the authors duly pointed out, there exist a body of literature that examines the potential of field-margin flower plantings to enhance beneficial arthropod conservation and ecosystem service delivery on agricultural lands. However, I agree with the authors that more studies on the potential utility of flower plantings in extensively modified, large-scale agricultural landscapes are still needed. The present manuscript thus serves the purpose of filling this data gap. I also found the manuscript to be well written and well organized overall, and the ideas clearly communicated. I do have some issues that I believe need to be adequately addressed before the study could be published. None constitute fatal flaws in my opinion, and most of my more major concerns pertain to clarification of methodological and statistical approaches. My comments are as follows, hopefully they are helpful to the authors as they revise the manuscript.

Major / general comments:

  1. I would assume, naively, that treatment (i.e., floral planting vs. control) and % floral cover should be quite highly correlated (see L181). Figure 3 seems to suggest that at least the maximum values of floral cover are much higher in the floral planting margins. However, I don’t see any indication that an analysis was performed to explicitly examine the effect of treatment on % floral cover. I expect that if the floral plantings were at all successful, there should be a multicollinearity issue in including both treatment and % floral cover in the model; however, the authors do not discuss this potential issue at all. Conversely, if multicollinearity is not an issue, it means that floral plantings actually failed to systematically increase % floral cover, which would be interesting as well and would pose a challenge for how the results are currently interpreted by the authors (i.e., with the assumption that floral plantings increased floral resources). I think it is necessary for this issue to be discussed explicitly and carefully.
    Floral plantings did lead to significantly increased floral cover (Analyses were conducted as part of a separate study). This may well pose an issue with multicollinearity. To account for this, the authors modified their analyses to separately compare the effects of treatment and floral cover in different models, and adjusted results and discussion section accordingly (although final results and conclusions were predominately unaffected).
  2. Related to the above, the authors also stated that they collected data on what plant species were blooming during each survey event, as well as the relative abundances of each species, but did not present any analyses of these data. I recognize that the paper is already long and the discussion quite rich without this component, but I do wonder if some of the results could be best explained by flower diversity or the presence of particular taxa (as the authors also speculated), so it would be great if the authors could at least make some use of this strength in their dataset.
    The authors did conduct analyses with floral traits, including floral richness and other broad categories like availability of nectar to pollinators or predators. Many of the results about changes in floral composition are in a separate manuscript, and the analyses that directly assessed how predators or pollinators were affected by this had mostly non-significant results (although floral richness was strongly correlated with increased pollinator abundance). Additionally, the experiment was not specifically designed to test the effects of these other traits, and the authors chose to stick to the main hypotheses that the experiment was designed to test for publication.
  3. Also related to the major issue #1: there appear to be both additive and interactive effects of treatment and % floral cover. I think it is very interesting that sometimes these factors act additively and sometimes interactively, and would be very interested in seeing the authors discuss these two distinct effects—of course, assuming that there are no issues with multicollinearity.
    Since the authors modified their analyses to separately compare the effects of treatment and floral cover in different models, this is no longer an issue
  4. I don’t see study year listed explicitly as a fixed or random effect; please explain. The authors stated that month nested within year is a random effect, but if I correctly recall how nested random effects work, the author’s reported nesting scheme treats the same month in different years as distinct (i.e., May 2017 is treated as distinct from May 2018 and May 2019), while not explicitly accounting for the influence of year (i.e., the model does not consider the fact that different study years may have different underlying conditions that act across all fields in all sampling dates). Also, if I correctly recall, the nesting scheme disregards any commonality shared across each month in distinct years by treating all months in all years as distinct entities, which I don’t think is the right approach, as one WOULD expect that May in all years should share some characteristics in common, and June across all years share other characteristics, and so on. It seems to me that month and year should be CROSSED random effects rather than NESTED random effects in order to properly partition variance.
    Study year was used as a fixed effect in analyses that were not included in the final manuscript. It was not included because the study was not designed to test this, and there were no significant effects of year. Month was nested within year because sampling date in each month could vary by ~1-2 weeks between years depending on weather, and the authors thought it may be more appropriate to consider each sampling date as separate while still assuming year could have an impact. Additionally, the authors did conduct analyses with month and year as crossed random effects, and while z values and significance varied slightly, it did not affect any of the outcomes that were reported. For these reasons, the current nested random effects were kept.
    However, the authors recognize the rationale behind conducting analyses with crossed random effects as stated above, and are more than happy to change the analyses if the editors and reviewers think it more appropriate.
  5. Tangentially related to #4 above, since the data structure allows, it may even be worth including month as a fixed effect (probably categorical) to examine how any of the effects detected varies with season. I wonder if some of the most interesting patterns may be masked by treating month simply as a random term, instead of taking a closer look at how dynamics may potentially shift month to month. For example, perhaps during certain months, control margins consistently have higher floral cover than floral planting margins, and then vice versa in other months, such that in the aggregate, these patterns are lost. Also it may be interesting and relevant to isolate months during which potato flowers are in bloom to examine pollinator activity within the crop fields. I would encourage the authors to consider how best to make use of the available temporal resolution in their data to make more sense of their findings.
    The authors did conduct analyses with month as a fixed effect to determine if effects on pollinators and predators varied during different times of year. Additionally, we did consider when potato flowers were in bloom, although potato blooms were not explicitly quantified in any way. There was no significant increase in pollinator or predator abundance in crops during the months when potatoes were generally in bloom There were few notable overall differences based on month, and only early June was significantly different than other months for insect abundance. Because of this, the authors did not include these analyses in the manuscript.
  6. The authors state in L123-124, “To minimize among-field climatic, soil, and surrounding landscape variability, floral margins were compared to control margins for the same field.” This sounds like some sort of paired design, and as far as I know, the mixed effects models framework does not allow for constraining comparisons only among field margins in the same field. Please clarify. I would also like to see sample sizes for how many floral planting vs. control field edges there are in each year, and also, if possible, some discussion on whether fields that had more flower margins saw any benefits compared to fields that had only one planted margin.
    The above sentence was meant to reflect the fact that field was used as a random effect to account for the use of many different fields that each had floral and control margins. To clarify this point, the sentence was moved to the section on data analysis and changed to “Field identity was included as a random effect to account for among-field climatic, soil, and surrounding landscape variability.” Additionally, added the numbers of floral and control margins for each year (7 and 9 in 2017, 10 and 8 un 2018, and 6 and 6 in 2019). There was no effect of the number of floral margins on pollinators or predators, and because this wasn’t explicitly designed for or tested, the authors did not include these analyses.

Minor / specific comments:

This is a well-written and well-constructed manuscript! There are, however, still a few errors scattered here and there (e.g., L15 “plantings” should be “planting”, L211 “taxa” should be “taxon”, L267 “Chysopidae” should be “Chrysopidae”, L399 has a garbled “This research was funded by We acknowledge the support and funding from”; just to name the ones I picked up), so I would encourage the authors to carefully double-check.
Changed to “Planting”, changed to “taxon”, changed to “Chrysopidae”, removed “We acknowledge the support and funding from”.

L58: I’m not quite sure how to interpret “creating more sustainable existing agroecosystems” since “creating” and “existing” seem to contradict each other in this sentence. Please clarify?
Changed to “Modifying existing agroecosystems to be more sustainable”

L80-82: The way this sentence is structured seems to suggest that studies have always found that field margin plantings indeed promote both pollinators and predators, and they always spill over into crops. Is that the case? If not the case, please provide counterexamples or caveats?
Floral plantings do not always promote pollinators and predators, and beneficial insects do not always spill over into adjacent crops. This is addressed at the start of the paragraph below

L86: If not on-farm in conventionally managed landscapes, does that mean that previous work have just been done in experimental field settings? If that’s the case, perhaps that should be made a little clearer in the segment above (see also my comments re: L80-82).
Most previous studies were conducted either in experimental fields, or examined small-scale floral plantings when established in existing agroecosystems. The latter includes diversified agricultural settings, organic settings, and commercial, conventional settings. Rather than state all of these options and their relative frequency, the authors chose to focus more on where research was lacking.

Figure 1a: If possible, please put the map of Minnesota as an inset; perhaps a little smaller so that surrounding states could be shown. The current “floating” map looks somewhat awkward and may be confusing for those not familiar with US geography.
As this is mostly aesthetic, the authors chose to leave the map as it was. The concern was shrinking the map enough to be able see surrounding states would make it less useful, and the additional context from nearby states would not add much to those unfamiliar with US geography.

L134-135: Please comment on where this 15 m transect is placed in the margin? I assume that, like any ecotone, there are edge effects.
Added “Starting in the center of each corner margin, and moving towards the crop”

L135-136: 4 quadrats of 1 square meter seems adequate for this sampling scale if floral cover is very uniform, but perhaps inadequate if floral cover is highly patchy (which is the case in many natural systems). Could you comment on the uniformity of floral cover / species composition? Perhaps simply examining the variation that exists among the 4 quadrats in the same margin would provide a good answer.
With a few exceptions, floral cover was mostly uniform, and (at least anecdotally), seemed representative of the margins. Species composition was more variable between quadrats, although still captured essentially all species present in the margins.

L137-138: Verified via website using digital imagery taken in the field? Or collected samples?
Verified using digital imagery taken in the field

L144-145: I’m not quite clear on whether there is one set of within-crop sampling per crop field (i.e., 1 per circular planting of potatoes) or one set per field margin (i.e., 4 per circular planting of potatoes). Please clarify.
One set per field margin. To clarify this, changed to “Pollinators were sampled in field margins and inside the potato crops adjacent to each field margin. Sampling occurred at four locations in the crop: the edge (0 meters), and 10, 30, and 50 meters into the crop.”

L146: Are the “pollinator transects” actually “transects”? It sounds like it’s just walking around collecting pollinators (nothing wrong with that). I generally see “transects” as formally laid out paths that have a start and end point and that are used consistently, which does not seem to apply to this situation (at least, as written). It would be helpful to know if the total distance covered is kept consistent, or only sampling time (e.g., if no pollinators are seen, the observer walks more quickly).
Changed to “pollinator survey” to reflect the time-based vs. distance-based sampling method. The time of each survey was based on active sampling time. That is, the timer was stopped as soon as a pollinator was netted, and was started again only after it had been transferred to the kill jar and the sampler began walking again. Added “A timer was started and kept running until a pollinator was captured. At this point, the timer was stopped until the pollinator was transferred to a kill jar, and restarted once the sampler began walking again” for clarity.

L147: Table 2 shows that there are Bombyliidae flies, which are bee flies, not hover flies.
Added “and bee flies”

L150-151: Similar to the above comment, it would be useful to know if the distance walked is kept constant.
The distance was not explicitly kept constant (although was approximately the same), and was instead measured by the number of sweeps.

L151-152: Why pitfall traps? While pitfall traps have been known to collect pollinators (there are a few published studies out there), pollinator trapping usually involves painted bowl traps. Again, nothing wrong with this approach (plus, the sampling is already done), but it would be good to see a justification / explanation for why the authors took this approach. If the pitfalls were primarily used for predators, but incidentally also captured pollinators, that should be more clearly stated.
Bowl traps were deployed throughout the study, however the data were not used as bowl traps can lead to biased results when comparing areas with different floral density. Pollinators caught in pitfalls were incidental. Added “incidental catch from pitfall traps” to clarify.

L187-189: There are tools (e.g., emmeans package in R) to run these multiple pairwise comparisons; they should be considered. Otherwise, the authors are encouraged to apply some sort of multiple comparison correction themselves.
Applied Bonferroni correction to pairwise comparisons. Added “and applying a Bonferroni correction” to methods section, and changed graphs where appropriate.

L199-200: Is location within crop also included alternatively as categorical factor and as continuous variable for predators, like it was for pollinators?
Added “(alternatively as a continuous variable and as a factor)” in the section on predators for clarity.

L202-203: I am alas unable to access the supplementary information files. However, I do want to know if model selection via delta-AIC has caused any random effects to be dropped from modeling?
No random effects ended up being dropped from the models.

L227-228: Looking at Figure 3b, I would interpret this finding to mean that increasing cover ONLY affects pollinator abundance in crops adjacent to control margins, and NOT in crops adjacent to flower margins. When there’s such a strong interaction, the main effect result probably can’t be trusted. You can verify by checking each treatment level on its own (i.e., separate GLMMs).
Removed this sentence in editing

Figure 2, Figure 3, Figure 5: There are some graphical issues here; please fix.
Adjusted figures to correct issues

L313: What are beetle banks?
Beetle banks are usually non-flowering plantings used to promote beetles in agroecosystems. Because the main pertinent information about beetle banks can be inferred, the authors did not include additional elaboration.

L325-326: These are excellent points. The points also touch on the discussion of whether these plantings simply draw in pollinators from the landscape to concentrate in an area of rich flowers (i.e., a behavioral response) or if they actually “support” pollinator populations by actually increasing reproductive output of pollinators drawn to the plantings. It will be helpful for the authors to discuss this point (not necessarily at length, though I think acknowledging this data gap is important). Same logic with L384 where the authors state that floral plantings “increase the overall abundance of both pollinators and predators”—given the spatial scale considered, and the fact that planted and unplanted field margins seem to be intermixed, it seems plausible that the difference in abundance simply reflects movement of individuals from an untreated margin into a floral planting margin, without any overall abundance gains at the landscape scale.
Added “Related to this point, although floral plantings did increase pollinator and predator abundance in field margins, our study did not determine that floral plantings actually increased populations of pollinators or predators, instead of simply concentrating them in floral margins. Further study about the efficacy of floral plantings to truly increase beneficial insect populations in commercial, conventionally managed agroecosystems would be an important next step based on our work.” to the discussion section.

L350-351: This discussion reminds me: were there any significant sources of floral resources in bloom within the potato crop fields throughout all of the survey times? I would imagine that the potato blooming time is brief relative to the whole field season. This seems to point towards separating out the time points of crop bloom and analyzing those time points separately from other points of in-crop pollinator sampling. And if in-crop pollinator sampling was done at a time when there is negligible floral resources available in the crop, this should also be explicitly discussed. In fact, a more complete discussion on what floral resource conditions are like in the crop field overall would be appreciated.
The authors did consider when potato flowers were in bloom, although potato blooms were not explicitly quantified in any way. Anecdotally, multiple potato crops were in bloom during July and August, with several individual fields blooming during both sampling points. However, because the authors did not measure blooms within potato crops and because few significant differences were observed when analyzing results by month, the authors did not a discussion of this in the manuscript.

L402-404: Isn’t this funding information more appropriate in the funding section immediately above?
It is, and has been moved to the above section.